# Approach to Pyrido[2,1-*b*][1,3]benzothiazol-1-ones via In Situ Generation of Acyl(1,3-benzothiazol-2-yl)ketenes by Thermolysis of Pyrrolo[2,1-*c*][1,4]benzothiazine-1,2,4-triones

**DOI:** 10.3390/molecules28145495

**Published:** 2023-07-18

**Authors:** Ekaterina A. Lystsova, Alexander S. Novikov, Maksim V. Dmitriev, Andrey N. Maslivets, Ekaterina E. Khramtsova

**Affiliations:** 1Department of Organic Chemistry, Perm State University, ul. Bukireva, 15, 614990 Perm, Russia; liscova_ea@mail.ru (E.A.L.); dmax@psu.ru (M.V.D.); koh2@psu.ru (A.N.M.); 2Institute of Chemistry, Saint Petersburg State University, Universitetskaya nab. 7/9, 199034 St. Petersburg, Russia; a.s.novikov@spbu.ru; 3Research Institute of Chemistry, Рeoples’ Friendship University of Russia (RUDN University), Miklukho-Maklaya Street, 6, 117198 Moscow, Russia

**Keywords:** acyl(imidoyl)ketene, cycloaddition, DFT calculations, heterocumulene, hetero-Diels–Alder reaction, thermolysis, thermal analysis

## Abstract

Acyl(imidoyl)ketenes are highly reactive heterocumulenes that enable diversity-oriented synthesis of various drug-like heterocycles. Such ketenes, bearing heterocyclic substituents, afford angularly fused pyridin-2(1*H*)-ones in their [4+2]-cyclodimerization reactions. We have utilized this property for the development of a new synthetic approach to pharmaceutically interesting pyrido[2,1-*b*][1,3]benzothiazol-1-ones via the [4+2]-cyclodimerization of acyl(1,3-benzothiazol-2-yl)ketenes generated in situ. The thermal behaviors of 3-aroylpyrrolo[2,1-*c*][1,4]benzothiazine-1,2,4-triones and 3-benzoylpyrrolo[2,1-*b*][1,3]benzothiazole-1,2-dione (two new types of [*e*]-fused 1*H*-pyrrole-2,3-diones reported by us recently) have been studied by thermal analysis and HPLC to elucidate their capability to be a source of acyl(1,3-benzothiazol-2-yl)ketenes. As a result, we have found that only 3-aroylpyrrolo[2,1-*c*][1,4]benzothiazine-1,2,4-triones are suitable for this. The experimental results are supplemented with computational studies that demonstrate that thermolysis of 3-aroylpyrrolo[2,1-*c*][1,4]benzothiazine-1,2,4-triones proceeds through an unprecedented cascade of two thermal decarbonylations. Based on these studies, we discovered a novel mode of thermal transformation of [*e*]-fused 1*H*-pyrrole-2,3-diones and developed a new pot, atom, and step economic synthetic approach to pyrido[2,1-*b*][1,3]benzothiazol-1-ones. The synthesized drug-like pyrido[2,1-b][1,3]benzothiazol-1-ones are of interest to pharmaceutics, since their close analogs show significant antiviral activity.

## 1. Introduction

Pyrido[2,1-*b*][1,3]benzothiazol-1-ones (Figure 1) are valuable scaffolds for medicinal chemistry. For example, their derivatives were found to be useful in the development of antiviral (anti-dengue [1], anti-flavivirus [2,3], inhibitors of NS5B polymerase of hepatitis C virus [4], antivirals against respiratory syncytial, influenza (Flu-V), and herpes simplex viruses [5]), antimicrobial (against Gram-positive bacteria (*Bacillus subtilis*, *Bacillus thuringiensis*) [6]), and anticancer (against HepG2 liver cancer cells [7] and Ehrlich-Lettre ascites carcinoma cells [8]) agents (Figure 1).

Acyl(imidoyl)ketenes are versatile synthetic platforms that enable diversity-oriented synthesis (DOS) of various drug-like heterocyclic systems on their basis [9]. These compounds are highly reactive chemical species and can only be used in organic synthesis if generated in situ [9]. Nevertheless, their reactions tend to have high yields, short reaction times, high selectivity, pot, atom and step economy (PASE), and simple purification procedures [9].

At the same time, acyl(imidoyl)ketenes **A** bearing heterocyclic substituents are known to afford angular heterocycles **B** or **C** in their [4+2]-cyclodimerization reactions (Figure 1) [9]. Obviously, heterocycles **C** have in their structures angularly fused pyridin-2(1*H*)-one moiety, which, in principle, can open up the possibility of synthesis of pyrido [2,1-*b*][1,3]benzothiazol-1-ones on their basis. At present, only one method of generation in situ of acyl(imidoyl)ketenes **A** bearing heterocyclic substituents is known, thermal decarbonylation of [*e*]-fused 1*H*-pyrrole-2,3-diones (FPDs) **D** (Figure 1) [9].

Recently, we have reported [10] a new type of FPDs, 3-aroylpyrrolo [2,1-*c*][1,4]benzothiazine-1,2,4-triones (APBTTs) **1**, prone to undergo a 1,4-thiazine ring contraction reaction to afford corresponding 1,3-thiazole derivatives under certain conditions [11]. These seem to be a promising feature for the development of a synthetic strategy to pyrido [2,1-*b*][1,3]benzothiazol-1-ones through chemical transformations of APBTTs **1**.

Considering the high pharmaceutical interest in pyrido [2,1-*b*][1,3]benzothiazol-1-ones, in this paper we report a new synthetic approach to novel derivatives of pyrido [2,1-*b*][1,3]benzothiazol-1-ones **2** via the [4+2]-cyclodimerization of acyl(1,3-benzothiazol-2-yl)ketenes **3** generated in situ from APBTTs **1** (Figure 2). The experimental results were supplemented with computational studies to elucidate the reaction mechanism.

## 2. Results and Discussion

For the development of the approach to pyrido [2,1-*b*][1,3]benzothiazol-1-ones **2**, we considered two types of potent sources of acyl(1,3-benzothiazol-2-yl)ketenes **3**, APBTTs **1** [10,11] and 3-benzoylpyrrolo [2,1-*b*][1,3]benzothiazole-1,2-dione **4** (Figure 3) [11]. Compound **4** was assumed to undergo a classical pattern for FPDs of thermal decarbonylation [9], cheletropic elimination of CO from C*^1^*=O position, affording the desired ketene **3a**. Compounds **1** were hypothesized to undergo elimination of two molecules of CO from C*^1^*=O (as a classical pattern of thermal decarbonylation for FPDs [9]) and C*^4^*=O positions (as a reactivity feature of APBTTs **1**, observed by us earlier in nucleophilic reactions [11]), affording the desired ketenes **3**.

To check our assumptions, we studied the thermal decomposition of compounds **1a–g**, **4** by simultaneous thermal analysis (STA) (Table 1, Figure 2 and Figure 3, Appendix A). According to the data obtained, APBTTs **1a**–**g** underwent thermal decomposition with a weight loss accompanied by an exothermic effect (Figure 2). The values of the weight loss corresponded to the elimination of two CO molecules from APBTTs **1a**–**g** (Table 1), which was in a good accordance with our assumptions about the thermolysis pattern of these compounds. In the case of compound **4**, it underwent thermal decomposition with a weight loss accompanied by an exothermic effect consisting of two peaks (Figure 3). The value of the weight loss was one and a half times more than the calculated one for the elimination of CO molecules from compound **4** (Table 1). Apparently, compound **4** took a different pathway of thermolysis from that proposed by us.

Examination by HPLC-UV (Appendix A) of the products of thermolysis of compounds **1a**–**g**, **4,** obtained by measuring their melting points in a capillary, revealed that thermolysis of APBTTs **1a**–**g** resulted in the formation of a single product, and thermolysis of compound **4** resulted in a complex mixture of products, among which only trace amounts of thermolysis product corresponding to APBTT **1a** were observed. In addition, compounds **1a**–**d** decomposed without melting (visually dark violet solid turned to a yellow solid), and compounds **1e**–**g**, **4** melted with decomposition.

Thus, the results of STA and HPLC-UV (Appendix A) clearly indicated that APBTTs **1a**–**g** could be suitable candidates for the synthesis of compounds **2a**–**g**, while compound **4** was not.

Then, we successfully scaled up the thermolysis of APBTTs **1a**–**g** under solvent-free conditions to 0.3 mmol (about 100 mg) and isolated products **2a**–**g** in good yields (58-91%) by simple recrystallization of the reaction mixtures. The structures of compounds **2a**,**f** were unequivocally proved by single crystal X-ray analyses (CCDC 2277018 (**2a**), 2277017 (**2f**), Appendix A). It should be mentioned that compounds **2a**–**g** had very poor solubility, and there were problems with the acquisition of their NMR spectra; therefore, their ^13^C NMR spectra were obtained involving solid-state (ssNMR) and cryoprobe NMR techniques (Appendix A).

Since the structures of compounds **2a**,**f** were proved by single crystal X-ray analyses, they can be considered as reference structures for the establishment of the structures of other compounds **2** by comparison of their spectral characteristics.

So, ^1^H NMR spectra of compounds **2** show a set of signals of aromatic protons, one of which has a segregate signal in the region of 9.51 ppm (for spectra recorded in CDCl_3_) or 9.35–9.31 ppm (for spectra recorded in DMSO-*d*_6_) (Figure 4, Section 3.2, Appendix A). This characteristic signal is produced by C*^9^*H moiety, which is deshielded by the carbonyl group C*^1^*=O of the pyridinone moiety. In ^13^C NMR spectra of compounds **2**, nine characteristic signals (with the exception of signals of substituents in aromatic rings) are segregated from a set of aromatic signals. In compound **2a**, they are the signal of the carbon atom of an aroyl carbonyl group C(Ar)=O at 191.8 ppm, the signal of the carbon atom of an ester carbonyl group OC(Ar)=O at 163.6 ppm, the signal of the carbon atom C*^1′^* of a 2-(1,3-benzothiazol-2-yl) substituent at 161.2 ppm, the signal of the carbon atom of a carbonyl group C*^1^*=O at 158.3 ppm, the signal of the carbon atom of a pyridinone moiety C*^4a^* at 156.0 ppm, the signal of the carbon atom of a pyridinone moiety C*^3^* at 155.2 ppm, the signal of the carbon atom C*^3a′^* of a 2-(1,3-benzothiazol-2-yl) substituent at 152.0 ppm, and two signals of carbon atoms of a pyridinone moiety C*^2^* and C*^4^* at 108.6 and 108.7 ppm (Figure 4, Section 3.2, Appendix A). NMR spectra of compounds **2b**–**g** show similar characteristic signals (Section 3.2, Appendix A). In IR spectra of compounds **2**, there are two characteristic bands. One of them corresponds to vibrations of an ester carbonyl group OC(Ar)=O at 1735–1756 cm^−1^, and the other one, a joint band, corresponds to vibrations of a ketone aroyl carbonyl group C(Ar)=O and a lactam carbonyl group C*^1^*=O of the pyridinone moiety at 1654–1665 cm^−1^ (Section 3.2, Appendix A).

To elucidate the possible mechanism of the investigated reaction, we performed computational DFT studies (Appendix A).

So, transformation of APBTT **1a** to pyrido [2,1-*b*][1,3]benzothiazol-1-one **2a** could proceed through several pathways, which are different combinations of two decarbonylations, [4+2]-cyclodimerization, and 1,3-acylotropic shift (Figure 4). At the initial stage, APBTT **1a** could undergo two patterns of decarbonylation, a classical variant of elimination of C*^1^*=O to afford ketene **I1** [9], and an unusual variant of elimination of C*^4^*=O to give 3-benzoylpyrrolo [2,1-*b*][1,3]benzothiazole-1,2-dione **4**. Then, intermediates **I1**, **4** could decarbonylate again to result in ketene **3a**, which could undergo a [4+2]-cyclodimerization→1,3-acylotropic shift cascade**,** or intermediate **I1** could undergo a classical pathway of [4+2]-cyclodimerization with a subsequent 1,3-acylotropic shift→decarbonylation or decarbonylation→1,3-acylotropic shift cascade.

We were most interested in the first stage of the process under study, since experimentally we observed that thermolysis of compound **4** did not afford the target compound **2a** (results of STA and HPLC-UV studies).

Results of DFT calculations of total electronic energies, enthalpies, and Gibbs free energies of reaction for elementary stages of different pathways for **1a** → **2a** transformation (Table 2) revealed that transformation **1a** → **3a** occurred via intermediate **4** (an unprecedented pathway), whereas the formation of alternative intermediate **I1** was found to be highly thermodynamically unfavorable. So, according to these calculations, **1a** → **2a** transformation proceeded through **1a** → **4** → **3a** → **I2** → **2a** sequence.

Note that results of DFT calculations of total electronic energies, enthalpies, and Gibbs free energies of activation for elementary stages of different pathways for **1a** → **3a** transformation (Figure 5, Table 3) also revealed that transformation **1a** → **3a** occurred via intermediate **4**, whereas the formation of alternative intermediate **I1** was found to be highly thermodynamically and kinetically unfavorable.

Thus, we observed a discrepancy between theoretical calculations and experimental observations. So, keeping the obtained results in mind, we suppose that when individual compound **4** was heated slowly and gradually from room temperature in our experiments, it underwent another way of decomposition from that we assumed above (Figure 3). This makes thermolytic transformations of 3-benzoylpyrrolo [2,1-*b*][1,3]benzothiazole-1,2-dione **4** an intriguing object for further studies.

## 3. Materials and Methods

### 3.1. General Information

^1^H and ^13^C NMR spectra (Appendix A) were acquired on a Bruker Avance III 400 HD spectrometer (Bruker BioSpin AG, Faellanden, Switzerland) (at 400 and 100 MHz, respectively) or on a Bruker Avance 500 Neo (Bruker BioSpin AG, Faellanden, Switzerland) equipped with a Prodigy (BBO) broadband cryoprobe (at 125 MHz for ^13^C nuclei) in CDCl_3_ (stab. with Ag) or DMSO-*d*_6_ using solvent residual signals (in ^13^C NMR, 77.00 for CDCl_3_, 39.51 for DMSO-*d*_6_; in ^1^H NMR, 7.26 for CDCl_3_, 2.50 for DMSO-*d*_6_) as internal standards. ^19^F NMR spectrum (Appendix A) was acquired on a Bruker Avance III 400 HD spectrometer (Bruker BioSpin AG, Faellanden, Switzerland) (at 376 MHz) in DMSO-*d*_6_ using C_6_H_5_CF_3_ signals as internal standard. ^13^C ssNMR spectra (Appendix A) were acquired on a Bruker Avance III 400 WB NMR spectrometer (Bruker BioSpin AG, Faellanden, Switzerland) (at 100 MHz). IR spectra were recorded on a Perkin–Elmer Spectrum Two spectrometer (PerkinElmer Inc., Waltham, MA, USA) from mulls in mineral oil. Melting points were measured on a Mettler Toledo MP70 apparatus (Mettler-Toledo (MTADA), Schwerzenbach, Switzerland). Elemental analyses were carried out on a Vario MICRO Cube analyzer (Elementar Analysensysteme GmbH, Langenselbold, Germany). The reaction conditions were optimized using HPLC-UV [Hitachi Chromaster (Hitachi High-Technologies Corporation, Tokyo, Japan); NUCLEODUR C18 Gravity column (particle size 3 μm; eluent acetonitrile–water, flow rate 1.5 mL/min); Hitachi Chromaster 5430 diode array detector (λ 210–750 nm)]. STA (Appendix A) was performed using a Netzsch STA 449 F1 Jupiter (Netzsch - Gerätebau GmbH, Wittelsbacherstraße, Germany) with temperature programs 10 K/min under an argon atmosphere (20 mL/min). The single crystal X-ray analyses of compounds **2a**,**f** were performed on an Xcalibur Ruby diffractometer (Agilent Technologies) (Oxfordshire, UK). The empirical absorption correction was introduced by multi-scan method using SCALE3 ABSPACK algorithm [12]. Using OLEX2 [13], the structures were solved with the SHELXS [14] program and refined by the full-matrix least-squares minimization in the anisotropic approximation for all non-hydrogen atoms with the SHELXL [15] program. Hydrogen atoms were positioned geometrically and refined using a riding model. HPLC-UV chromatograms were analyzed in MultiChrom program for Windows [16]. Thin-layer chromatography (TLC) was performed on Merck silica gel 60 F_254_ plates using EtOAc/toluene, 1:5 *v*/*v*, toluene, and EtOAc as eluents. Starting compounds **1a**–**g**, **4** were obtained according to reported procedures [10,11]. All other solvents and reagents were purchased from commercial vendors and used as received. Procedures involving compounds **1a**–**g**, **4** were carried out in oven-dried glassware.

#### General Procedure for Compounds **2a**–**g**

The corresponding APBTT **1** (0.3 mmol) was put into an oven-dried test tube and pressed slightly. Then, it was heated at 220 °C on a metal bath for 3 min. The reaction mixture was cooled to room temperature and scrubbed with toluene (5 mL). The resulting precipitate was filtered off and recrystallized from toluene to afford an appropriate pyrido [2,1-*b*][1,3]benzothiazol-1-one **2**.
*2-(1,3-Benzothiazol-2-yl)-4-benzoyl-1-oxo-1H-pyrido [2,1-b][1,3]benzothiazol-3-yl benzoate* (**2a**). Yield: 71 mg (85%); yellow solid; mp 299–301 °C. ^1^H NMR (400 MHz, CDCl_3_): δ = 9.51 (m, 1 H), 7.90 (m, 2 H), 7.68–7.48 (m, 8 H), 7.37–7.13 (m, 6 H), 7.00 (m, 1 H) ppm. ^13^C NMR (100 MHz, CDCl_3_): δ = 191.8, 163.6, 161.2, 158.3, 156.0, 155.2, 152.0, 140.4, 137.5, 135.4, 133.2, 131.1, 130.3 (2C), 129.1, 128.8, 128.2 (2C), 127.8 (2C), 127.8, 127.7 (2C), 127.6, 125.6, 124.8, 122.4, 121.9, 121.0 (2C), 108.7, 108.6 ppm. IR (mineral oil): 1742, 1657 cm^−1^. Anal. Calcd (%) for 3C_32_H_18_N_2_O_4_S_2_ · C_7_H_7_: C 69.97; H 3.53; N 4.75. Found: C 69.68; H 3.43; N 5.18. The crystal structure of compound **2a** was deposited at the Cambridge Crystallographic Data Centre with the deposition number CCDC 2277018. *Crystal Data*: C_32_H_18_N_2_O_4_S_2_, *M* = 558.60, triclinic, *a* = 7.2896(13) Å, *b* = 16.634(3) Å, *c* = 21.579(2) Å, α = 80.604(11)°, β = 88.828(11)°, γ = 85.776(14)°, *V* = 2574.4(7) Å^3^, *T* = 295(2), space group *P*–1, *Z* = 2, μ(MoKα) = 0.251 mm^−1^. The final refinement parameters: *R*_1_ = 0.0842 [for observed 5887 reflections with *I* > 2σ(*I*)], *wR*_2_ = 0.2491 (for all independent 12279 reflections, *R*_int_ = 0.0619), *S* = 1.059. Largest diff. peak and hole, 0.434 and −0.531 ēÅ^−3^.*2-(1,3-Benzothiazol-2-yl)-4-(4-methylbenzoyl)-1-oxo-1H-pyrido [2,1-b][1,3]benzothiazol-3-yl 4-methylbenzoate* (**2b**). Yield: 88 mg (89%); yellow solid; mp 311–313 °C. ^1^H NMR (400 MHz, CDCl_3_): δ = 9.51 (m, 1 H), 8.04–7.85 (m, 3 H), 7.72–7.44 (m, 7 H), 7.34–7.01 (m, 5 H), 2.44 (s, 3 H), 2.17 (s, 3 H) ppm. ^13^C NMR (125 MHz, CDCl_3_): δ = 191.7, 163.6, 161.2, 158.4, 156.0, 155.0, 152.0, 143.9, 141.9, 137.4, 135.3, 130.4 (2C), 129.0, 128.8 (2C), 128.7, 128.4 (2C), 128.0 (2C), 127.7, 127.5, 126.2, 125.5, 125.3, 124.7, 122.5, 121.9, 121.0, 120.9, 108.8, 21.8, 21.3 ppm. IR (mineral oil): 1742, 1654 cm^−1^. Anal. Calcd (%) for C_34_H_22_N_2_O_4_S_2_: C 69.61; H 3.78; N 4.78. Found: C 70.00; H 3.73; N 4.64.*2-(1,3-Benzothiazol-2-yl)-4-(4-chlorobenzoyl)-1-oxo-1H-pyrido [2,1-b][1,3]benzothiazol-3-yl 4-chlorobenzoate* (**2c**). Yield: 94 mg (81%); yellow solid; mp 312–314 °C. ^1^H NMR (400 MHz, CDCl_3_): δ = 9.51 (m, 1 H), 8.07 (m, 2 H), 7.92 (m, 2 H), 7.73–7.08 (m, 11 H) ppm. ^13^C ssNMR (100 MHz): δ = 191.6, 161.8, 159.5, 153.7, 150.1, 138.3, 136.5, 132.8, 129.5, 126.2, 122.5, 118.6, 110.5 ppm. IR (mineral oil): 1751, 1663 cm^−1^. Anal. Calcd (%) for C_32_H_16_Cl_2_N_2_O_4_S_2_: C 61.25; H 2.57; N 4.46. Found: C 61.39; H 2.63; N 4.29.*2-(1,3-Benzothiazol-2-yl)-4-(4-bromobenzoyl)-1-oxo-1H-pyrido [2,1-b][1,3]benzothiazol-3-yl 4-bromobenzoate* (**2d**). Yield: 97 mg (91%); yellow solid; mp 318–320 °C. ^1^H NMR (400 MHz, CDCl_3_): δ = 9.51 (m, 1 H), 8.00–7.89 (m, 4 H), 7.79–7.30 (m, 10 H), 7.12 (m, 1 H) ppm. ^13^C ssNMR (100 MHz): δ = 191.6, 162.1, 159.5, 154.4, 150.2, 138.7, 136.6, 132.8, 129.7, 126.3, 122.8, 121.0, 118.7, 110.5, 107.2 ppm. IR (mineral oil): 1752, 1665 cm^−1^. Anal. Calcd (%) for C_32_H_16_Br_2_N_2_O_4_S_2_: C 53.65; H 2.25; N 3.91. Found: C 53.31; H 2.08; N 3.77.*2-(1,3-Benzothiazol-2-yl)-4-(4-fluorobenzoyl)-1-oxo-1H-pyrido [2,1-b][1,3]benzothiazol-3-yl 4-fluorobenzoate* (**2e**). Yield: 65 mg (73%); yellow solid; mp 283–285 °C. ^1^H NMR (400 MHz, DMSO-*d*_6_): δ = 9.35 (m, 1 H), 8.30 (m, 1 H), 8.11 (m, 1 H), 7.82–7.67 (m, 5 H), 7.61–7.43 (m, 1 H), 7.39–7.28 (m, 4 H), 7.19–7.12 (m, 2 H), 6.96 (m, 1 H) ppm. ^13^C NMR (125 MHz, CDCl_3_): δ = 190.2, 167.0, 165.5, 165.0, 163.5, 162.5, 161.0, 158.1, 155.5, 155.3, 151.9, 137.3, 136.4, 136.4, 135.2, 132.9, 132.8, 130.4, 130.3, 128.6, 127.9, 127.7, 125.7, 125.1, 125.1, 124.9, 122.3, 122.0, 121.0, 120.9, 115.4, 115.4, 115.2, 115.2, 108.5, 108.3 ppm. ^19^F NMR (376 MHz, DMSO-*d*_6_): δ = -107.18, -110.75 ppm. IR (mineral oil): 1747, 1661 cm^−1^. Anal. Calcd (%) for C_32_H_16_F_2_N_2_O_4_S_2_: C 64.64; H 2.71; N 4.71. Found: C 64.52; H 2.73; N 4.65.*2-(1,3-Benzothiazol-2-yl)-4-(furan-2-carbonyl)-1-oxo-1H-pyrido [2,1-b][1,3]benzothiazol-3-yl furan-2-carboxylate* (**2f**). Yield: 47 mg (58%); yellow solid; mp 272–274 °C. ^1^H NMR (400 MHz, DMSO-*d*_6_): δ = 9.31 (m, 1 H), 8.23 (m, 1 H), 8.13 (m, 1 H), 8.08 (m, 1 H), 8.01 (m, 1 H), 7.77–7.69 (m, 2 H), 7.51 (m, 1 H), 7.40 (m, 2 H), 7.34 (m, 1 H), 7.13 (m, 1 H), 6.81 (m, 1 H), 6.69 (m, 1 H) ppm. ^13^C NMR (125 MHz, CDCl_3_): δ = 177.5, 160.9, 154.8, 154.5, 154.0, 152.2, 151.9, 147.0, 146.8, 144.3, 137.5, 136.5, 135.3, 128.1, 127.8, 127.6, 125.7, 125.0, 124.9, 122.4, 121.8, 121.1, 120.9, 119.6, 119.3, 112.2, 112.2, 108.4 ppm. IR (mineral oil): 1756, 1660 cm^−1^. Anal. Calcd (%) for C_28_H_14_N_2_O_6_S_2_: C 62.45; H 2.62; N 5.20. Found: C 62.59; H 2.43; N 5.28. The crystal structure of compound **2f** was deposited at the Cambridge Crystallographic Data Centre with the deposition number CCDC 2277017. *Crystal Data*: 2(C_28_H_14_N_2_O_6_S_2_)∙C_7_H_8_, *M* = 1169.20, triclinic, *a* = 9.4943(16) Å, *b* = 12.3983(17) Å, *c* = 12.9210(16) Å, α = 111.584(13)°, β = 92.997(14)°, γ = 107.840(15)°, *V* = 1322.7(4) Å^3^, *T* = 295(2), space group *P*–1, *Z* = 1, μ(MoKα) = 0.253 mm^−1^. The final refinement parameters: *R*_1_ = 0.0579 [for observed 3726 reflections with *I* > 2σ(*I*)], *wR*_2_ = 0.1505 (for all independent 6240 reflections, *R*_int_ = 0.0481), *S* = 1.035. Largest diff. peak and hole, 0.234 and −0.274 ēÅ^−3^.*2-(1,3-Benzothiazol-2-yl)-4-(thiophene-2-carbonyl)-1-oxo-1H-pyrido [2,1-b][1,3]benzothiazol-3-yl thiophene-2-carboxylate* (**2g**). Yield: 60 mg (70%); yellow solid; mp 278–280 °C. ^1^H NMR (400 MHz, DMSO-*d*_6_): δ = 9.33 (m, 1 H), 8.22 (m, 1 H), 8.12 (m, 1 H), 8.06 (m, 1 H), 8.00 (m, 1 H), 7.92 (m, 1 H), 7.78–7.69 (m, 3 H), 7.40–7.33 (m, 2 H), 7.25 (m, 1 H), 7.20 (m, 1 H), 7.05 (m, 1 H) ppm. ^13^C NMR (125 MHz, CDCl_3_): δ = 182.6, 160.9, 158.7, 158.2, 154.2, 154.1, 152.0, 143.1, 137.6, 135.3, 135.0, 134.1, 133.8, 133.7, 132.5, 128.1, 127.8, 127.7, 127.6, 127.5, 125.7, 124.8, 122.4, 121.8, 121.0, 121.0, 109.2, 108.8 ppm. IR (mineral oil): 1735, 1655 cm^−1^. Anal. Calcd (%) for C_28_H_14_N_2_O_4_S_4_: C 58.93; H 2.47; N 4.91. Found: C 58.67; H 2.04; N 4.82.

### 3.2. Computational Details

The DFT calculations for all model structures were carried out at the M06-2X/6-31G* level of theory with the help of the Gaussian-09 program package [17]. No symmetry restrictions have been applied during the geometry optimization procedure. The Hessian matrices were calculated analytically for all optimized model structures to prove the location of the correct minimum or saddle point (transition state) on the potential energy surface. The Cartesian atomic coordinates for all model structures are presented in attached xyz-files (Appendix A).

## 4. Conclusions

The thermal decomposition of 3-aroylpyrrolo [2,1-*c*][1,4]benzothiazine-1,2,4-triones **1** and 3-benzoylpyrrolo [2,1-*b*][1,3]benzothiazole-1,2-dione **4** was investigated to elucidate their ability to generate acyl(1,3-benzothiazol-2-yl)ketenes **3**. Based on these results, a series of novel pyrido [2,1-*b*][1,3]benzothiazol-1-ones **2** was prepared from compounds **1** in a high yield of up to 91%. According to computational studies, the thermolysis of compounds **1** proceeded via an unprecedented cascade of two thermal decarbonylations 1→4→acyl(1,3-benzothiazol-2-yl)ketenes **3**. The developed pyrido [2,1-*b*][1,3]benzothiazol-1-ones **2** are of interest to medicine and pharmaceutics.

## Data Availability

The presented data are available in this article.

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
