# Peer review of "Approach to Pyrido[2,1-b][1,3]benzothiazol-1-ones via In Situ Generation of Acyl(1,3-benzothiazol-2-yl)ketenes by Thermolysis of Pyrrolo[2,1-c][1,4]benzothiazine-1,2,4-triones"

_molecules, 2023, doi:10.3390/molecules28145495_

Round 1

Reviewer 1 Report

Ekaterina A. Lystsova et al. described the synthesis of a series of pyrido[2,1-][1,3]benzothiazol-1-ones via the [4+2]-cyclodimerization of in situ-generated acyl(1,3-benzothiazol-2-yl)ketenes s by thermolysis of Pyrrolo[2,1-c][1,4]benzothiazine-1,2,4-triones, whose reaction course was computationally rationalized to proceed through an unprecedented cascade of two thermal decarbonylations. The study has relevant results and is interesting for readers. However, the manuscript has some issues to address before further consideration.:

ü  Abstract: The final sentence of the abstract did not correctly finalize the outlook of this study. In fact, it is out of context since it seems to be a justification rather than a conclusion. Therefore, I recommend writing a better conclusive sentence about the study’s aim and scope.

ü  Linse 103 and 100: HPLC-UV chromatograms must be added to the supplementary material and appropriately cited.

ü  Line 115: Explanations about the structure deduced by the single crystal X-ray analysis must be provided.

ü  Line 117 and others: The existing NMR spectra in the supplementary material should be cited.

ü  Line 118: Clear explanations about the relevant NMR signals leading to compound identification must be provided in this section.

ü  Lines 156-159: This important sentence must be improved since it is challenging to be followed. In other words, this explanation is crucial for the process under interest and must be rewritten to add better clarity.

ü  The language style of R&D section must be improved. I could deduce the process and aim of the explanations, but it requires improvement since some passages are incorrectly described/explained and, therefore, are challenging to follow.

ü  The discussion is poorly developed since the R&D section is highly descriptive.

ü  Conclusions summarize the results and should contain conceptual findings from a mechanistic point of view. Indeed, the last sentence is not a conclusion related to the study’s results.

The manuscript requires language editing since several grammar and stylistic issues are found throughout the manuscript. Several passages are challenging to be followed.

Author Response

  1. The abstract was revised, and the required conclusive sentence was added.
  2. HPLC-UV chromatograms were added to SI and cited in the main manuscript.
  3. Single crystal X-ray analyses data for compounds 2a,f (CCDC 2277018 (2a), 2277017 (2f)) were deposited to CCDC in accordance with Instructions for Authors in Molecules. These data are available free of charge. Discussion on single crystal X-ray analyses is not necessary in this paper since this information will be redundant in this paper (the present manuscript does not cover crystallographic characteristics of the synthesized compounds). Such discussion insertions can be considered as an unjustified increase in the size of the article.
  4. A citation on Supplementary Materials was added.
  5. The discussion on it was added to the text.
  6. The sentence was revised.
  7. The text was revised and improved.
  8. We added some additional discussions in this section, and now we hope that it is acceptable.
  9. The last sentence aims to emphasize the potential practical interest in the synthesized compounds.

Reviewer 2 Report

It would be better in the Supplementary Information Section to present the 1H and 13C NMR spectra in the landscape format

Author Response

Actually, we do not know what is «landscape format» for 1H and 13C NMR spectra. In the present submission, 1H and 13C NMR spectra are provided in accordance with NMR Guidelines for ACS Journals recommended by Instructions for Authors in Molecules.

Reviewer 3 Report

E. E. Khramtsova and her group described a synthesis of pyrido[2,1-b][1,3]benzothiazol-1-ones via hetero-Diels-Alder of acylketenes. It is interesting highly reactive acylketenes were generated from heterocyclic diones 4 and/or 1 via a loss of CO, and the ketenes gave [4+2] cycloaddition products along with 1,3-acyl shift. The resulting two products were deposited by X-ray crystal diffraction and their reaction mechanism could also be supported by DFT calculations.

The text, synthetic procedure, and English are fine, but some structural analysis is not enough to approve the exact structures. So, I strongly recommend that all structures make clear by fine NMR spectra or XRD.

Author Response

Single crystal X-ray analyses data for compounds 2a,f (CCDC 2277018 (2a), 2277017 (2f)) were deposited to CCDC in accordance with Instructions for Authors in Molecules.

Compounds 2a-g show similar spectral data, which means they are structurally alike and are of the same series. Structures of two representatives of this series were unambiguously proved by single crystal X-ray analyses (so they can be considered as reference structures), which is sufficient to confirm the structure of all compounds of the series.

Supporting information to the present manuscript contains plots of NMR spectra for all new synthesized compounds. The text description of spectral data for all new synthesized compounds is available in the full text of the manuscript. Their single crystal X-ray analyses are available free of charge in CCDC.

We also added a short discussion on spectral data and structure elucidation to the main text of the manuscript.

Round 2

Reviewer 3 Report

I understand that the X-ray spectra of compounds 2a and 2f were deposited at CCDC.
However, for example, the 1H NMR spectrum of 2a is contaminated with a co-solvent (toluene?), and it has interfered with the assignment by 13C NMR. Can't you use toluene-d8 for the NMR solvent?
13C NMR of 2f is also insufficient to discriminate the signal.
If the authors did not measure X-ray analysis, they have to attach clear NMR charts of 2a and 2f to ensure other compounds' structures.

Author Response

Yes, 1H NMR spectrum of 2a is contaminated with a co-solvent (Yes, it is toluene. We purified compound 2a by recrystallization from toluene). This is not a contaminant in this case, just compound 2a forms very stable solvates with toluene. And it is irresistibly hard to remove toluene from it.

The NMR signals of toluene in deuterated solvents are well known (doi: 10.1021/jo971176v) and they can be readily separated from the signals of compound 2a. In the case of 13C NMR of compound 2a, there is no overlapping of signals of toluene and compound 2a.

The usage of toluene-d8 as NMR solvent will never solve this problem, since

  1. toluene-d8 contains residual signals of partially deuterated toluene which will appear in 1H NMR;
  2. In 13C NMR, toluene-d8 will overlap and hide all the aromatic signals of compound 2a;
  3. the only option of how toluene-d8 can help with NMRs is that we use it as a solvent for recrystallization of compound 2a. But in these case, the consumption of this expensive solvent will be very great, and I am not sure that such experiment will provide an excellent result, since the compound 2a will still contain toluene-d8 as a solvate, which will show additional signals in 1H and 13C NMR spectra.

NMR charts of 2a and 2f are already present in Supplementary Materials to the present submission, along with NMR charts of all compounds from the series 2. We added a discussion on structure elucidation and characteristic signals in spectra for compounds 2 to the main manuscript. We also added additional data on X-ray analyses to section 3 and Supplementary Materials.